# Understanding Optimization Challenges
# when Encoding to Geometric Structures

**Babak Esmaeili**[*]                                            B.ESMAEILI@UVA.NL
*Amsterdam Machine Learning Lab,*
*University of Amsterdam*

**Robin Walters**[*]                                       R.WALTERS@NORTHEASTERN.EDU
*Khoury College of Computer Sciences,*
*Northeastern University*

**Heiko Zimmermann**                                      H.ZIMMERMANN@UVA.NL
*Amsterdam Machine Learning Lab,*
*University of Amsterdam*

**Jan-Willem van de Meent**                               J.W.VANDEMEENT@UVA.NL
*Amsterdam Machine Learning Lab,*
*University of Amsterdam*

**Editors:** Sophia Sanborn, Christian Shewmake, Simone Azeglio, Arianna Di Bernardo, Nina Miolane

## Abstract

Geometric inductive biases such as spatial curvature, factorizability, or equivariance have been shown to enable learning of latent spaces which better reflect the structure of data and perform better on downstream tasks. Training such models, however, can be a challenging task due to the topological constraints imposed by encoding to such structures. In this paper, we theoretically and empirically characterize obstructions to training autoencoders with geometric latent spaces. These include issues such as singularity (e.g. self-intersection), incorrect degree or winding number, and non-isometric homeomorphic embedding. We propose a method, isometric autoencoder, to improve the stability of training and convergence to an isometric mapping in geometric latent spaces. We perform an empirical evaluation of this method over 2 domains, which demonstrates that our approach can better circumvent the identified optimization problems.

**Keywords:** Representation Learning, Autoencoders, Homeomorphism, Topological, Equivariant, Lie Groups, Isometry

## 1. Introduction

Recent research has shown that geometric inductive biases can be helpful for representation learning (Higgins et al., 2022, 2018; Pfau et al., 2020). Representations that are equivariant to transformations of the underlying generative factors, many of which can be elegantly described by geometric priors such as groups (rotation, translation, etc.), make it easier to reason about the similarities of different instances in the dataset. In this paper, we focus on geometric inductive biases in the context of unsupervised and weakly-supervised models for representation learning. While the prevailing intuition is that an inductive bias that matches the underlying topology of the data will guide a model towards a useful representation, there

---

[*] Equal contribution

Esmaeili Walters* Zimmermann van de Meent

are also indications that certain inductive biases can make a model more difficult to train in practice (Park et al., 2022; Falorsi et al., 2018).

To understand why encoding to geometric structures can give rise to optimization challenges, we formalize different topological defects that can occur in a randomly initialized encoder, such as discrepancies in the winding number or crossing number relative to those in a homeomorphic encoder. We show that these topological defects will be preserved under the assumption of continuous optimization, which shows that escaping these local optima relies on the discrete jumps that are employed during optimization (Section 3). Moreover, we show that, even in the absence of topological defects, a homeomorphic encoder may violate isometry, which is to say that equidistant points on the abstract manifold may not map to equidistant points in the latent space. To alleviate these challenges to optimization, we propose an autoencoder objective (Section 4) that encourages equidistant points on the abstract manifold to map to equidistant points in latent space. Experiments demonstrate that this objective can help to escape local optima during the early stages of training, resulting in more reliable convergence and a greater degree of isometry after training.

## 2. Problem Statement

**Homeomorphic Autoencoders.** We focus on learning representations in domains where we can associate a known geometry with input data. Concretely, we assume that our data lies on a low-dimensional manifold $\mathcal{M}$ that is embedded in a higher-dimensional input space $\mathcal{X} := \mathbb{R}^n$ via a mapping $g_{\mathcal{X}} : \mathcal{M} \to \mathcal{X}$. This is commonly known as the *manifold hypothesis* (Bengio et al., 2013). We denote the image of the mapping by $\mathcal{M}_x := g_{\mathcal{X}}(\mathcal{M}) \subseteq \mathcal{X}$. Then $g_{\mathcal{X}}$ is a homeomorphism, or topological isomorphism, onto its image $g \colon \mathcal{M} \xrightarrow{\sim} \mathcal{M}_x$. That is, $g$ is continuous, bijective, and has a continuous inverse.

Our goal is to learn a mapping $f_\phi \colon \mathcal{X} \to \mathcal{Z}$ for some suitable latent space $\mathcal{Z}$ such that the encoder is itself a homeomorphism when restricted to the embedded manifold $\mathcal{M}_x$. To do so, we follow the design proposed by Falorsi et al. (2018) $f_\phi \colon \mathcal{X} \xrightarrow{h_\phi} \mathcal{Y} \xrightarrow{\pi} \mathcal{Z}$. Here $\mathcal{Y} := \mathbb{R}^d$ is an intermediate Euclidean space capable of embedding $\mathcal{M}$. The first function $h_\phi$ is an ordinary neural network. The second function $\pi$ is parameter-less mapping which can be interpreted as a projection to $\mathcal{Z}$. For example, For example, $\mathcal{Y} := \mathbb{R}^2$ and $\pi$ is the Gram-Schmidt process in the case of group SO(2). As we assume we know the geometry of the data, we set $\mathcal{Z}$ to be $\mathcal{M}$

To learn a homeomorphic encoder $f_\phi|_{\mathcal{M}_x} \colon \mathcal{M}_x \to \mathcal{M}$ in an unsupervised manner, we will introduce a decoder network $f_\theta^\dagger \colon \mathcal{Z} \to \mathcal{X}$. The composition $f_\theta^\dagger \circ f_\phi$ then defines an autoencoder that can be trained by minimizing the reconstruction loss $\mathcal{L}_{\phi,\theta}(x) := d_{\mathcal{X}}(x, \hat{x})$ based on a distance metric $d_{\mathcal{X}}$, where $\hat{x} := f_\theta^\dagger(f_\phi(x))$. Note that while minimizing the reconstruction loss ensures that $f_\theta^\dagger \circ f_\phi|_{\mathcal{M}_x}$ approximates the identity, it need not be the case that the composition $\eta_\phi = f_\phi \circ g \colon \mathcal{M} \to \mathcal{M}$ is also the identity. Rather it is a self-bijection on $\mathcal{M}$ representing the different potential choices of parameterizations of $\mathcal{M}$.

## 3. Theory

In this paper, we uncover topological obstructions that are distinct from the homological obstruction identified by de Haan and Falorsi (2018). The main insight that we put forward

is that imposing a geometric structure on the latent space introduce additional obstructions *during optimization*, which we call optimization obstructions. We now discuss several specific optimization obstructions. We focus on the case where $\mathcal{M}$ is the Lie group SO(2) with underlying manifold $S^1$, $y \in \mathbb{R}^2$, and $z = \pi(y) = y/\|y\|$. All the optimization obstructions we consider occur in this case and in the case of higher-dimensional manifolds $\mathcal{M}$ as well, but are simpler to describe for $S^1$. See Appendix B & E for more details and proofs.

### 3.1. Figure Eight Local Minima

To understand what obstructions might arise during optimization, we consider continuous-time training along a gradient flow. We denote the weights of the initialized encoder as $\phi(0)$ and the trained weights as $\phi(1)$. We consider the idealized setting in which $\phi(t)$ is a continuous function of $t$. That is, $f_{\phi(t)}(x)$ is a homotopy from the initialized encoder $f_{\phi(0)}(x)$ to the trained one $f_{\phi(1)}(x)$. Empirically, either at initialization or after some training, we often observe a "figure 8" pattern in $\mathcal{Y}$ (Figure 11, bottom left). Once this local minimum is obtained, it is very difficult to move out of it in practice using gradient descent. We make this intuition precise by observing that continuous optimization preserves the ordering of points on the circle as demonstrated by Proposition 1 and Figure 13.

**Proposition 1** *Assume that $f_{\phi(t)}$ undergoes continuous optimization. Assume that $f_{\phi(t)} \circ g$ is injective for all t. The cyclic ordering induced on k points by $f_{\phi(0)}$ is equal to $f_{\phi(1)}$. Thus a figure 8 embedding, cannot be transformed to a homeomorphic embedding.*

In other words, transition from a "figure 8" to a homeomorphic embedding is impossible without violating continuity during optimization. This suggests that when training with SGD, we will need to rely on the stochasticity of the gradient estimate and discrete jumps to escape this local optimum.

### 3.2. Degree Obstructions

A second obstruction is that continuous optimization preserves the winding number of a mapping. This number is defined as follows. If the embedded image $h_\phi(\mathcal{M}_x)$ does not contain the origin, then the mapping factors through $\mathbb{R}^2 \setminus \{(0,0)\}$ and is consequently continuous. In this case $\eta_\phi = \pi \circ h_\phi \circ g$ has a well-defined degree, also known as winding number $w(\eta_\phi) \in \mathbb{Z}$. In order to be a homeomorphic embedding the winding number must be $w(\eta_\phi) \in \{-1, 1\}$. Under random initialization, however, the initial network may have winding number equal to any integer. Assuming continuous optimization then $h_{\phi(t)}(x)$ is a continuous function of both $x, t$. We assume that $h_{\phi(t)}(x) \neq (0,0)$ for any $t, x$ and thus winding number is defined for any time. The following proposition thus holds.

**Proposition 2** *The winding number of the initialized model and final model are equal* $w(\pi \circ h_{\phi(0)} \circ g) = w(\pi \circ h_{\phi(1)} \circ g)$.

In practice, neither the continuous optimization assumption nor the avoidance of the origin holds. Rather $h_\phi$ is updated by SGD in discrete jumps and $h_\phi \circ g$ may map to the origin. Thus, empirically, we do see that if the weights and learning rate is high enough, the winding number may change. For more discussion on this, see Appendix B.

Esmaeili Walters* Zimmermann van de Meent

| | L-shaped Tetrominoe | | | Teapot | | | Airplane | |
|---|---|---|---|---|---|---|---|---|
| | # H. | Isometry | # H. | Isometry | # H. | Isometry | | |
| AE | 0/15 | $141.28 \pm 39.$ | 2/15 | $114.68 \pm 66.$ | 0/15 | $104.94 \pm 44.$ | | |
| DAE | 0/15 | $133.83 \pm 26.$ | 2/15 | $117.22 \pm 71.$ | 0/15 | $91.40 \pm 31.$ | | |
| Isom-AE | **12/15** | **$44.94\pm 79.$** | **12/15** | **$24.48\pm43.$** | **9/15** | **$80.43\pm105.$** | | |
| Isom-AE + $y$-reg | 9/15 | $82.71 \pm 104.20$ | 8/15 | $74.51 \pm 81.$ | 5/15 | $124.10 \pm 93.53$ | | |
| Sup-AE | **15/15** | **$5.74\pm0.49$** | **13/15** | **$5.67\pm 0.$** | 0/15 | $96.78 \pm 38.$ | | |

Table 1: Evaluation of latent space for different objectives trained on different images.

## 4. Isometric Autoencoders

We argue that preserving relative distance is key for a representation to be useful in most downstream tasks. For example, if we want to cluster our latent representations based on there orientation, we want the the distances in latent space to reflect the actual distance in the abstract manifold $\mathcal{M}$. For the purpose of representation learning, we do not necessary need the encodings to match the ground truth exactly as long as we are able to *compare* instances in a meaningful way. We can formalize this concept in terms *isometry*. A function $\eta : \mathcal{X} \to \mathcal{Z}$ is isometric if $d_{\mathcal{X}}(x_1, x_2) = d_{\mathcal{Z}}(\eta(x_1), \eta(x_2))$ for all $x_1, x_2 \in \mathcal{X}$. We now define our notion of optimality based on this property.

**Definition 3** *Let $g, f_\phi, \eta_\phi$ be defined as above. The encoder $f_\phi$ is optimal if $\eta_\phi$ is isometric.*

While an isometry is desirable, it is generally almost impossible to guarantee and practically difficult to verify for a function with neural network components. Therefore, we need a notion of isometry that is both computable and achievable using standard gradient descent optimization methods. Motivated by these practical limitations, we define a less restrictive property that can be verified in practice in Appendix A. Moreover, we propose an objective that encourages to learn an optimal encoder by making slightly stronger assumptions about the data.

Concretely, we assume our dataset has the form $\mathcal{D}\{\{x_{i,1} \ldots x_{i,K}\}\}_{i=1}^{N}$ generated as follows. We assume that $\mathcal{M}$ is a group manifold, for example SO(2). We then sample $m_1, m_T \in \mathcal{M}$. We iteratively apply the transformation $m_T$ to get a sequence $m_k = m_T m_{k-1} \in \mathcal{M}$. Then we pass $m_k$ to the ground truth generative network $x_k = g(m_k)$. Let $z_k = f(x_k)$ be the encoding of $x_k$ and $z_{(k-1,k)} = z_k z_{k-1}^{-1}$ be the group element that sends $z_{k-1}$ to $z_k$. In each sequence, we know $z_{(k-1,k)}$ should be identical for all $k$. We define our objective as

$$\mathcal{L}_{\phi,\theta}^{\text{Isom-AE}}(\{x_k\}_{k=1}^{K}) = \sum_{k=1}^{K} d_{\mathcal{X}}(x_k, \hat{x}_k) + \alpha \sum_{k=2}^{K-1} d_{\mathcal{Z}}(z_{(k-1,k)}, z_{(k,k+1)}), \tag{1}$$

## 5. Experiments

We perform a series of experiments to explore the difficulty of learning an optimal $f_\phi$. Concretely, we investigate how often do we fall into one of the failure cases described in Section 3. We also examine whether training with the Isom-AE objective helps in terms of learning an optimal encoder. The baselines are discussed in Appendix D.

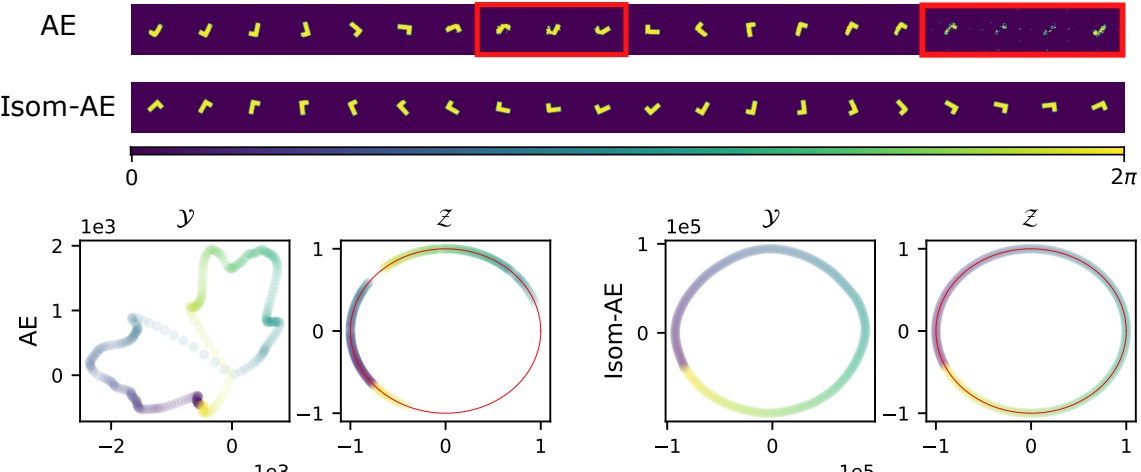

Figure 1: Latent space interpolation in the decoder (*Top*) and encoder (*Bottom*).

**Evaluation.** We evaluate all models based on three criteria (1) homeomorphism, (2) isomtery, and (3) reconstructions. Similar to isometry, it is practically difficult to determine if a learned mapping is homeomorphic. Here, we verify homeomorphism by examining the encoder to check if it has (1) winding number 1 or -1, (2) crossing number 0, (3) yields a continuous path when interpolating in the data manifold from $-\pi$ to $\pi$. For evaluating isometry, we simply compute and report the term $L_p(N)$ described in Definition 4 for $p = 2$ and $N = 10000$ rather than applying a binary threshold.

**Images.** In this experiment, we train on images of an L-shaped tetromino, a teapot, and an airplane (Shilane et al., 2004). The SO(2) manifold corresponding to each object is made by rotating the image of the object around the center. We describe the full sequence generation process in Appendix H.

**Results.** We report our findings in Table 1. Similar to the 3D crown experiment, we observe that even though the types of obstructions vary across images, both AE and DAE in general fail to learn a homeomorphic encoder. The Isom-AE objective improves performance noticeably across all metrics. Contrary to the 3D-Crown experiment (Appendix D), we observe that $y$ regularization is not very helpful. We observed the main failure case for most of the non-homeomorphic encoders was due to discontinuity. Isom-AE resolved this issue for the most part, which allows us to interpolate nicely in the $\mathcal{Z}$-space (Figure 1). What is very surprising is that in the case of airplane, we see that even supervised objective fails to overcome these optimizations obstructions, while an Isom-AE is able to achieve this nicely.

## 6. Conclusion

In this paper, we investigate obstructions to optimization encountered when learning encoders to topological spaces. This work contains several limitations. Firstly, our theoretical analysis is limited by the idealized assumptions necessary to analyze the method using topological tools which do not exactly match those encountered in practice. Secondly, the metrics we define such isometry and harder to define and compute for higher dimensional manifolds. Future work includes expanding our analysis and techniques to a boarder array of Lie groups and non-group manifolds.

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

ESMAEILI WALTERS* ZIMMERMANN VAN DE MEENT

# Appendix for "Understanding Optimization Challenges when Encoding to Geometric Structures"

## Appendix A. Isometric Representations

Even when a learned representation is homeomorphic, it may not be isometric, in the sense that equidistant points on the abstract manifold $\mathcal{M}$ may not map to equidistant points in the latent space $\mathcal{Z}$. As an example, let us consider the case where $\mathcal{M}$ is the special orthogonal Lie group SO(2) defined as

$$\text{SO}(2) = \{z \mid z \in \text{GL}(2), z^T z = I, \det(z) = 1\} = \left\{ A(y) := \begin{bmatrix} y_1 & -y_2 \\ y_2 & y_1 \end{bmatrix} \mid y \in \mathbb{R}^2, \|y\| = 1 \right\}, \tag{2}$$

where GL(2) is the general linear group, the group of invertible $2 \times 2$ matrices under matrix multiplication. Define, for example, the embedding $g \colon \text{SO}(2) \to \mathbb{R}^2$ by $z \mapsto z(0,2)^\top$, embedding the manifold as a circle of radius 2. We consider $f_\phi$ mapping this embedding back to the group by first encoding to $\mathcal{Y} \subseteq \mathbb{R}^2$ via $h_\phi$ and then projecting to the circle $S^1$, the manifold underlying the Lie group, via $\pi \colon \mathbb{R}^2 \to \text{SO}(2)$ defined $y \mapsto A(y/\|y\|)$.

We can construct a mapping $f_\phi = \pi \circ h_\phi$ by first mapping $x \in \mathbb{R}^2$ to points on an ellipse,

$$h_\phi(x) := (a\cos\theta(x), b\sin\theta(x))^\top, \qquad \theta(x) = \text{atan2}(x_2, x_1), \tag{3}$$

and then projecting them to SO(2) via $\pi$ as described above. Restricted to $\mathcal{M}_x := g(\text{SO}(2))$, $f_\phi$ is an homeomorphism. Moreover, as demonstrated in Figure 15 the resulting mapping $\eta_\phi$ might not preserve the distances between instances on the manifold $\mathcal{M}$ when mapping to the latent space $\mathcal{Z}$.

### A.1. Approximate Isometry

While an isometry is desirable, it is generally almost impossible to guarantee and practically difficult to verify for a function with neural network components. Therefore, we need a notion of isometry that is both computable and achievable using standard gradient descent optimization methods. Motivated by these practical limitations, we define a less restrictive property that can be verified in practice:

**Definition 4** *Let $f : \mathcal{X} \to \mathcal{Y}$. Let $x_i, x_i' \sim p_{\mathcal{X}}(x)$ for $1 \leq i \leq N$, $y_i = f(x_i)$, $y_i' = f(x_i')$, and $q_i = d_{\mathcal{X}}(x_i, x_i')/d_{\mathcal{Y}}(y_i, y_i')$. Let $\mathbf{1} = (1, \ldots, 1) \in \mathbb{R}^n$. Define*

$$L_p(N) = \|q - \mathbf{1}\|_p. \tag{4}$$

*We say $f$ is* approximately-isometric with probability greater than $1 - \alpha$ *(for $\epsilon, \alpha, p, N$), if*

$$\mathbb{P}(L_p(N) < \epsilon) > 1 - \alpha. \tag{5}$$

## Appendix B. Theory: Additional Details

### B.1. Proposition for the figure "8" shape

Let $(z_1, z_2, z_3, z_4)$ denote the four end points of the two disjoint intervals of $f_{\phi(0)}$. Since there is no starting point for the circle, an ordering of these points is only well defined up to cyclic permutations in the group $C_4$, that is $(z_1, z_2, z_3, z_4) \mod C_4 = (z_2, z_3, z_4, z_1) \mod C_4$. Proposition 1 now states that continuous optimization must preserve an ordering up to equivalence. The proof is illustrated in Figure 13.

### B.2. Winding number in practice

Empirically, we do see that if the weights and learning rate is high enough, the winding number may change. However, if the initialization avoids origin, then due to the tendency of the magnitude of the unnormalized embeddings $h_\phi(x)$ to grow during optimization, the winding number changing becomes more unlikely and thus a significant obstruction to finding homeomorphic embeddings under optimization. Winding number is also the primary optimization obstruction which makes it impractical to remove the hard projection $\pi$. If instead we decode directly from $y \in \mathcal{Y}$ but push embeddings to the unit circle using the loss $|\|y\| - 1|$, then it is far more likely we converge to discontinuous embeddings with the incorrect winding number (Appendix G).

### B.3. Magnitude Growth in $\mathcal{Y}$

Empirically, we observe the values of the embeddings in $\mathcal{Y} = \mathbb{R}^2$ continually grow during training. This phenomenon makes it more difficult for the embedded data manifold $h_\phi(\mathcal{M}_x)$ to cross the origin and for the winding number to change. We give a theoretical explanation for this behavior.

Consider what would happen if the embedding $y$ were updated directly based on the gradient of the loss $\nabla_y \mathcal{L}$ with respect to $y$. We assume the loss depends only on $z = y/\|y\|$, and so has level sets which are unions of radial rays from the origin. The gradient $\nabla_y \mathcal{L}$ must then be tangent to a circle about the origin. That is, for $y = (a, b)$, the gradient $\nabla_y \mathcal{L} = (\pm b, \mp a)$. Under gradient flow, the evolution of $y$ in time $y_t$ would thus flow along circles of fixed radius and so $\|y_0\| = \|y_t\|$. Under gradient descent, however, due to the convexity of the flow lines, which are circular, the embeddings $y$ will tend to grow in magnitude. For $\eta \in \mathbb{R}_{>0}$, we compute

$$\|y - \eta \nabla_y \mathcal{L}\|^2 = (a \mp \eta b)^2 + (b \pm \eta a)^2 = (a^2 + b^2)(1 + \eta^2) > \|y\|^2.$$

In practice, however, we do not update $y$ based on $\nabla_y \mathcal{L}$ but rather based on the gradient with respect to model parameters $\phi$. Let $F \colon \Phi \to \mathcal{Y}$ be the map from model parameters $\phi$ to $y$ given fixed input data $x$. Then the actual update to $y$ is $\tilde{\nabla}_y \mathcal{L} = dF^T \circ \nabla_\phi \mathcal{L}$ where $dF$ is the total derivative or Jacobian of the map $F$. Since $\nabla_\phi \mathcal{L} = (dF)\nabla_y \mathcal{L}$ we have $\tilde{\nabla}_y \mathcal{L} = dF^T dF \nabla_y \mathcal{L}$. The angle between $\tilde{\nabla}_y \mathcal{L}$ and $\nabla_y \mathcal{L}$ is bounded by some $\theta$ a quantity depending on the eigenvalues of the operator $dF$. Given that $\nabla_y \mathcal{L}$ is tangential to the circle, assuming for simplicity $\tilde{\nabla}_y \mathcal{L}$ has constant length $L$ and uniform distribution $[-\theta, \theta]$ in angle to $\nabla_y \mathcal{L}$, the norm of $y$ still grows *in expectation*.

**Proposition 5** *Assume a circle of radius $R$. Let $v$ be a random vector at $y$ on the circle of length $L < R$ with angle to the tangent uniform in $[-\theta, \theta]$. Then*

$$\mathbb{E}[\|y + v\|^2] = \frac{\phi\left(L^2 + 2R^2\right) + L\sin(\phi)(2 - L\cos(\phi))}{2\phi} > \|y\|^2.$$

## Appendix C. Related Work

**Unsupervised or Weakly Supervised Learning of Geometric Representations** There has been a large amount of work concerned with learning representations from data with geometric structure in the unsupervised, weakly supervised, or semi-supervised setting. To this end, different methodologies have been brought forward, including the use of hyperspherical prior distributions (Davidson et al., 2018a), specializations of the reparameterization trick (Falorsi et al., 2018; Rey et al., 2019), and making use of local connectivity information (Moor et al., 2020; Chen et al., 2021; Lee et al., 2021).

**Learning Disentangled Representations.** Topological group structure in data can also be used to define a notion of disentanglement based on the invariance properties of the representations under certain group transformations (Higgins et al., 2018). This has given rise to a corpus of work that aims to learn disentangled group representation in various settings (Caselles-Dupré et al., 2019; Quessard et al., 2020; Tonnaer et al., 2022; Pfau et al., 2020; Zhu et al., 2021). Tonnaer et al. (2022) propose an objective which is similar our Isom-AE objective in Section 1 to learn Linear Symmetry-Based Disentangled (LSBD) representations. However, this objective assumes knowledge of ground truth group elements while our objective only assumes that sequences are generated by repeated application of the same unknown group element. Our work differs from these approaches in that we are not interested in disentanglement, but focus on fundamental topological obstructions when encoding to group with non-trivial topological structure.

**Topological Obstructions in Learning** To develop a better understanding of the failure modes that we observe in Figure 14, we will formalize topological obstructions to training homeomorphic embeddings in Section 3. Homological obstructions to auto encoding have been observed (de Haan and Falorsi, 2018; Batson et al., 2021; Falorsi et al., 2018; Rey et al., 2019) in previous work. de Haan and Falorsi 2018, Theorem 1 give a formal condition. For any latent space $\mathcal{Z}$ with non-trivial topology, it is possible to learn an encoder $f_\phi$ that is continuous when restricted to $\mathcal{M}_x \subset \mathcal{X}$, but this encoder must be discontinuous on the full space $\mathcal{X}$. For this reason, Falorsi et al. (2018) and others (Xu and Durrett, 2018; Davidson et al., 2018b; Meng et al., 2019) use the two-part encoder like ours, inserting discontinuous layers $\pi$ when mapping to circles, spheres, $SO(n)$, or other manifolds. This explicit discontinuity circumvents the homological obstruction without forcing the linear layers of the network to approximate discontinuities using large weights, which we and others find leads to instability during training and inferior reconstructions (Section 5 and Falorsi et al. (2018)).

## Appendix D. Experimental Details

**Baselines.** Throughout our experiments, we compare against (1) a standard AE where the objective is only based on reconstruction loss, (2) a supervised AE where in addition to minimizing the reconstruction loss, the encoder is trained to predict the ground truth representations. These two scenarios serve as extremes on the spectrum of guiding the model to the right representation. As another baseline, we also compare against a denoising autoencoders (DAE) which aids to learn a smoother embedding space. In addition to Isom-AE, we also tried regularizing the y-space to be close $S_1$ (by penalizing $(\|y\|_2 - 1)^2$) in order to mitigate the optimization problem discussed in Subsection B.3, which we refer to as "Isom-AE + reg-$y$".

We train all our models for 100 epochs with a batch-size of 100. For optimization, we use the RAdam optimizer (Liu et al., 2019) with a learning rate of 5e-4. For low-dimensional cases, we use a 4-layer MLP with 512 hidden units followed by a tanh activation for both the encoder and decoder architecture (Table 2). For the high-dimensional image datasets, we use a 4-layer CNN with kernel, stride, and padding of size 4, 2 and 1 respectively followed by a *leakyReLU* activation (Table 3). All models were initialized and trained with 15[1] different random seeds. Though we hypothesize that these topological defects may occur in a variety of topological structures, in this paper we restrict our experiments to the Lie group $SO(2)$ with $\pi$ defined as in Section 2.

| Encoder |
| --- |
| Input $x \in \mathbb{R}^3$ |
| F.C. 512, Tanh. |
| F.C. 512, Tanh. |
| F.C. 512, Tanh. |
| F.C. 2, $\pi(y) := y/\|y\|$. |

| Decoder |
| --- |
| Input $z \in \mathbb{R}^2$ s.t. $\|z\|_2 = 1$ |
| F.C. 512, Tanh. |
| F.C. 512, Tanh. |
| F.C. 512, Tanh. |
| F.C. 3. |

Table 2: Architecture of the Encoders and Decoders for the 3D crown dataset.

| Encoder |
| --- |
| Input $32 \times 32$ images |
| $4 \times 4$ conv. 32 stride 2, LeakyReLU. |
| $4 \times 4$ conv. 32 stride 2, LeakyReLU. |
| $4 \times 4$ conv. 64 stride 2, LeakyReLU. |
| $4 \times 4$ conv. 64 stride 2, LeakyReLU. |
| F.C. 2, $\pi(y) := y/\|y\|$. |

| Decoder |
| --- |
| Input $z \in \mathbb{R}^2$ s.t. $\|z\|_2 = 1$ |
| $4 \times 4$ deconv. 64, stride 2, LeakyReLU. |
| $4 \times 4$ deconv. 64, stride 2, LeakyReLU. |
| $4 \times 4$ deconv. 32, stride 2, LeakyReLU. |
| $4 \times 4$ deconv. 3, stride 2, LeakyReLU. |

Table 3: Architecture of the Encoders and Decoders for the image datasets.

---

1. We used a higher number of random seeds than normal to account for the training instability.

ESMAEILI WALTERS* ZIMMERMANN VAN DE MEENT

## Appendix E.  Proofs

We include the proofs for the propositions in the main text.

### E.1.  Figure Eight Local Minimum

**Proposition 6** *Assume that $f_{\phi(t)}$ undergoes continuous optimization and is thus continuous in $t$. Assume that $\pi \circ f_{\phi(t)} \circ g$ is injective for all $t$. The cyclic ordering induced on $k$ points by $f_{\phi(0)}$ is equal to that induced by $f_{\phi(1)}$. Thus a figure 8 embedding, which corresponds to cyclic order $(z_1, z_2, z_4, z_3) \bmod C_4$, cannot be transformed to a homeomorphic embedding, which has cyclic order $(z_1, z_2, z_3, z_4) \bmod C_4$ or $(z_4, z_3, z_2, z_1) \bmod C_4$.*

**Proof** Since we assume $\pi \circ f_{\phi(t)} \circ g$ is injective for all $t$, the path $\mathbf{z}(t) = (\pi \circ f_{\phi(0)} \circ g(\theta_i))_{i=1}^4$ is inside the $k$-fold configuration space on $S^1$ defined $\mathrm{Conf}_k(S^1) = \{(z_1, \ldots, z_k) \in (S^1)^k : z_i \neq z_j \text{ for } i \neq j\}$. In order to prove the claim, we will show that the path-connected components of $\mathrm{Conf}_k(S^1)$ correspond to cyclic orderings of $(z_1, \ldots, z_k)$ and thus the start and end point of every path share a cyclic ordering.

Mapping $(z_1, \ldots, z_k) \mapsto (z_k, (z_k^{-1} z_1, \ldots, z_k^{-1} z_{k-1}))$ gives a homeomorphism $\mathrm{Conf}_k(S^1) \cong \mathrm{SO}(2) \times \mathrm{Conf}_{k-1}(S^1 \setminus \{1\}) \cong \mathrm{SO}(2) \times \mathrm{Conf}_{k-1}(\mathbb{R})$. Let $\tilde{z}_i = z_k^{-1} z_i$. Consider $D = \{(\tilde{z}_1, \ldots, \tilde{z}_{k-1}) : \tilde{z}_1 < \ldots < \tilde{z}_{k-1}\} \subset \mathrm{Conf}_{k-1}(\mathbb{R})$.

We can identify the connected components of $\mathrm{Conf}_{k-1}(\mathbb{R})$. The set $D$ is a fundamental domain for the action of the symmetric group $S_{k-1}$ on $\mathrm{Conf}_{k-1}(\mathbb{R})$. Thus $\mathrm{Conf}_{k-1}(\mathbb{R}) = \coprod_{\sigma \in S_k} \sigma(D)$ is a disjoint union. Linear interpolation shows $D$ is connected. The sets $D$ and $\sigma(D)$ for $\sigma \in S_k$ are not connected. Consider a path from $\mathbf{z} = (z_1, \ldots, z_k) \in D$ to $\sigma(\mathbf{z}) \in \sigma(D)$. The element $\sigma$ must reverse the order of at least two elements $z_j < z_i$. Thus the function $f(\mathbf{z}) = z_i - z_j$ must take the value $0$ over the path by intermediate value theorem. Hence the path cannot be in $\mathrm{Conf}_{k-1}(\mathbb{R})$. Thus the connected components of $\pi_0(\mathrm{Conf}_{k-1}(\mathbb{R})) \cong S_{k-1}$.

Since $\mathrm{SO}(2)$ is connected, $\pi_0(\mathrm{Conf}_k(S^1)) \cong S_{k-1}$. That is each connected component of $\mathrm{Conf}_k(S^1)$ is labeled by an element of $S_{k-1}$ describing the ordering of $\tilde{z}_1, \ldots, \tilde{z}_{k-1}$ in $\mathbb{R}$. Each ordering of $(\tilde{z}_1, \ldots, \tilde{z}_{k-1})$ in turn corresponds to a different cyclic ordering of $z_1, \ldots, z_k$ in $S^1$, that is, a different element of $S_k/C_k$. Thus two $k$-point configurations are homotopic if and only if they have the same cyclic ordering. ∎

### E.2.  Degree Obstruction

**Proposition 7** *The winding number of the initialized model and final model are equal $w(\pi' \circ h_{\phi(0)} \circ g) = w(\pi' \circ h_{\phi(1)} \circ g)$.*

**Proof** The winding number of a map is a continuous function $t \mapsto w(\pi' \circ h_{\phi(t)} \circ g)$. Since the output space $\mathbb{Z}$ is discreet, the winding number must be constant in $t$. ∎

### E.3.  Magnitude Growth in $\mathcal{Y}$

As noted in the main text, we have $\tilde{\nabla}_y \mathcal{L} = (dF^T dF) \nabla_y \mathcal{L}$. *We assume that $dF$ is full rank, which is a reasonable assumption for an overparameterized neural network. In that case*

$M = dF^T dF$ is a positive definite symmetric matrix and can be orthogonally diagonalized $M = Q\Lambda Q^T$ where $Q$ is orthogonal and

$$\Lambda = \begin{pmatrix} \lambda_1 & 0 \\ 0 & \lambda_2 \end{pmatrix}$$

and $\lambda_i > 0$. The maximum angle between $\boldsymbol{x} = \nabla_y \mathcal{L}$ and $M\boldsymbol{x} = \tilde{\nabla}_y \mathcal{L}$ can then computed in terms of the eigenvalues $\lambda_i$. This maximum is computed for the case of an $n \times n$ symmetric positive definite matrix here[2]. We include the proof for the $2 \times 2$ case we consider here for completeness.

**Lemma 8** *The maximum angle between $x$ and $Mx$ for $x \in \mathbb{R}^2_{\neq 0}$ is*

$$\cos^{-1}\left(\frac{2\sqrt{\lambda_1 \lambda_2}}{\lambda_1 + \lambda_2}\right).$$

**Proof** The angle is maximized at the minimum value of

$$\frac{x^T M x}{\|x\|\|Mx\|}.$$

It suffices to consider $\|x\| = 1$. Substituting $M = Q\Lambda Q^T$ and $y = Qx$, we want to minimize

$$\frac{x^T Q^T \Lambda Q x}{x^T Q^T \Lambda^2 Q x} = \frac{y^T \Lambda y}{y^T \Lambda^2 y}$$

over all $\|y\| = 1$ since $\|Qx\| = \|x\| = 1$. Letting $a = y_1^2$ and noting $y_1^2 + y_2^2 = 1$, this is equal to minimizing

$$\frac{a\lambda_1 + (1-a)\lambda_2}{a\lambda_1^2 + (1-a)\lambda_2^2}$$

over $0 \le a \le 1$. Setting the derivative equal to 0 gives

$$\frac{(\lambda_1 - \lambda_2)^2(-\lambda_2 + a(\lambda_1 + \lambda_2))}{2\left(\lambda_2^2 + a(\lambda_1^2 - \lambda_2^2)\right)^{3/2}} = 0$$

and yields one critical value at $a = \lambda_2/(\lambda_1 + \lambda_2)$ corresponding to value $\frac{2\sqrt{\lambda_1 \lambda_2}}{\lambda_1 + \lambda_2}$. This is the global minimum since the boundary values $a = 0$ and $a = 1$ correspond to maxima with value 1. $\blacksquare$

Thus the angle between $\nabla_y \mathcal{L}$ and $\tilde{\nabla}_y \mathcal{L}$ is bounded by $\theta = \cos^{-1}\left(2\sqrt{\lambda_1 \lambda_2}/(\lambda_1 + \lambda_2)\right)$. Given that $\nabla_y \mathcal{L}$ is tangential to the circle, assuming for simplicity $\tilde{\nabla}_y \mathcal{L}$ has constant length $L$ and uniform distribution $[-\theta, \theta]$ in angle to $\nabla_y \mathcal{L}$, the norm of $y$ grows *in expectation*.

**Proposition 9** *Assume a circle of radius $R$. Let $v$ be a random vector at $y$ on the circle of length $L$ with angle to the tangent uniform in $[-\theta, \theta]$. Then*

$$\mathbb{E}[\|y + v\|^2] = L^2 + R^2 > R^2 = \|y\|^2.$$

2. karakusc (https://math.stackexchange.com/users/176950/karakusc), Maximum angle between a vector $x$ and its linear transformation $Ax$, URL (version: 2017-05-06): https://math.stackexchange.com/q/2266057

**Proof** Without loss of generality, $y = (0, R)$ and $v = (L \cos t, L \sin t)$ where $|t| < \theta$. Then we evaluate

$$
\begin{aligned}
\mathbb{E}[\|y + v\|^2] &= \frac{1}{2\theta} \int_{-\theta}^{\theta} \|(L \cos t, R + L \sin t)\|^2 \| dt \\
&= \frac{1}{2\theta} \int_{-\theta}^{\theta} (L^2 \cos^2 t + R^2 + 2RL \sin t + L^2 \sin^2 t) dt \\
&= \frac{1}{2\theta} \int_{-\theta}^{\theta} (L^2 + R^2) dt \\
&= L^2 + R^2.
\end{aligned}
$$

by Pythagorean identity and the fact $\sin t$ is odd. ∎

## Appendix F. Continuity Metric

For measuring continuity, we adopt a similar method as Falorsi et al. and evaluate continuity in terms of how the largest "jump" compare to others when walking a continuous path $m_i \in \mathcal{M}$ for $i = 1 \cdots N$ pairwise close points. We compute $q_i$ similar to Definition 4 by computing the relative distances

$$
q_i = \frac{d_{\mathcal{M}}(\eta_\phi(m_i), \eta_\phi(m_{i+1}))}{d_{\mathcal{M}}(m_i, m_{i+1})}.
$$

From the set $\{q_i\}_i$, we compute the continuity metric $L_{\text{cont}}$ as

$$
L_{\text{cont}} = \frac{M}{P_\alpha}, \qquad M = \max_i q_i, \qquad P_\alpha = \alpha\text{-th percentile of } \{q_i\}_{i=1}^N. \tag{6}
$$

In our experiments, we set $\alpha = 90$.

There are two differences between how we evaluate continuity compared to (Falorsi et al., 2018). First, we measure the continuity of $\eta_\phi$ rather than $f_\phi$ which we argue is more relevant in Section 4. Second, in (Falorsi et al., 2018), the authors are mainly interested in verifying whether the encoder is discontinuous in topological sense (which they verify by examining the inequality $M > \gamma P_\alpha$ for some $\gamma$). We on the other hand report continuity on the spectrum by computing the $\gamma$ that would make $\eta_\phi$ discontinuous.

## Appendix G. Decoding From $\mathcal{Y}$

We consider the alternate strategy of removing the projection $\pi$ and adding a loss so that $y$ stays close to the desired manifold $\mathcal{M}$ in $\mathcal{Y}$. Winding number obstructions become far more prominent in this case. In Figure 2, we show the latent space for different random seeds in the teapot case when we train with such an objective.

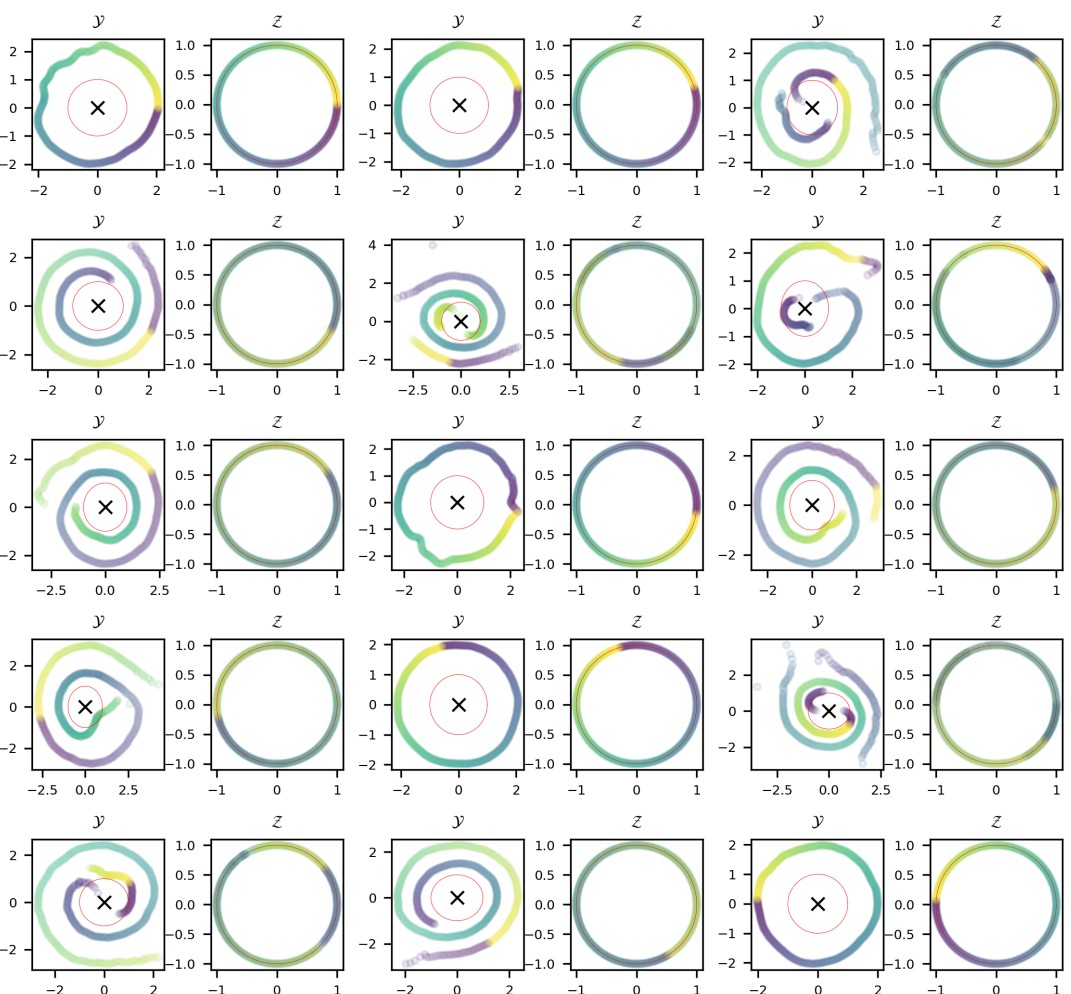

Figure 2: $\mathcal{Y}$ and $\mathcal{Z}$ space of Isom-AEs trained on teapots for 15 random seeds, where instead of decoding from $\mathcal{Z}$, we decode from $\mathcal{Y}$ with an additional soft regularization that constrain $y$ values to have unit length.

ESMAEILI WALTERS* ZIMMERMANN VAN DE MEENT

| | # H. | # W. | # C. | *3D-Crown* Cont. | Isometry | Recons. |
|---|---|---|---|---|---|---|
| AE | 6/15 | 14/15 | 9/15 | $97.56 \pm 139.53$ | $78.81 \pm 61.07$ | $\mathbf{0.00 \pm 0.01}$ |
| DAE | 7/15 | 14/15 | 8/15 | $171.72 \pm 260.73$ | $83.13 \pm 76.02$ | $\mathbf{0.00 \pm 0.00}$ |
| Isom-AE | 11/15 | 12/15 | 11/15 | $20.95 \pm 52.39$ | $48.21 \pm 106.02$ | $0.04 \pm 0.12$ |
| Isom-AE + reg-$y$ | $\mathbf{15/15}$ | $\mathbf{15/15}$ | $\mathbf{15/15}$ | $\mathbf{1.20 \pm 0.06}$ | $\mathbf{7.00 \pm 2.29}$ | $0.07 \pm 0.03$ |
| Sup-AE | $\mathbf{15/15}$ | $\mathbf{15/15}$ | $\mathbf{15/15}$ | $\mathbf{1.02 \pm 0.00}$ | $\mathbf{3.44 \pm 0.47}$ | $\mathbf{0.00 \pm 0.00}$ |

Table 4: Comparison of different AEs trained on the toy dataset in terms of number of encoders with homeomorphic mappings (# H.), correct winding number (# W.), and correct crossing number (# C.). We additionally the error on continuity, isometry, and reconstructions (lower is better).

## Appendix H. Additional Experiments

### H.1. 3D Crown

In our first experiment, we consider a simple dataset in $\mathbb{R}^3$ with parameterization $\mathbf{x} = (\cos\theta, \sin\theta, \sin 4\theta)$ for $\theta \in [-\pi, \pi]$. We generate 60k sequences of data points with length $K = 6$ In each sequence, a constant rotation of $\theta_i$ is applied where $\theta_i$ is sampled from $U(-\pi/5, \pi/5)$. For the objectives that do not rely on sequences, we simply flatten data before training.

**Results.** We report our findings in Table 4. We can see that in the majority of cases, both AE and DAE fail to learn a homeomorphic encoder. Interestingly, though the data is low-dimensional, it is still difficult to model due to non-linearity of the data manifold embedding. Concluding from the low crossing number (as well examining the latent spaces in Figure 10), the most common failure cases for learning a homeomorphic encoder are the figure eight obstruction as well as discontinuity. Isom-AE mitigates these problems to an extent, but not perfectly. We show an example of how isom-AE is able to unwind the latent the space to achieve a homeomorphic mapping in Figure 7[3]. Isom-AE with $y$ regularization, however, is able to avoid such obstructions completely, achieving a 100% homeomorphic success rate along with Sup-AE. We can also observe from Table 4 that homeomorphic mappings are overall more stable during training. Lastly, looking at the learning curves in Figure 5, we observe that winding number is susceptible to changing during training.

---

3. The $y$ values are very large, but we scale them down for visualizing near a unit circle.

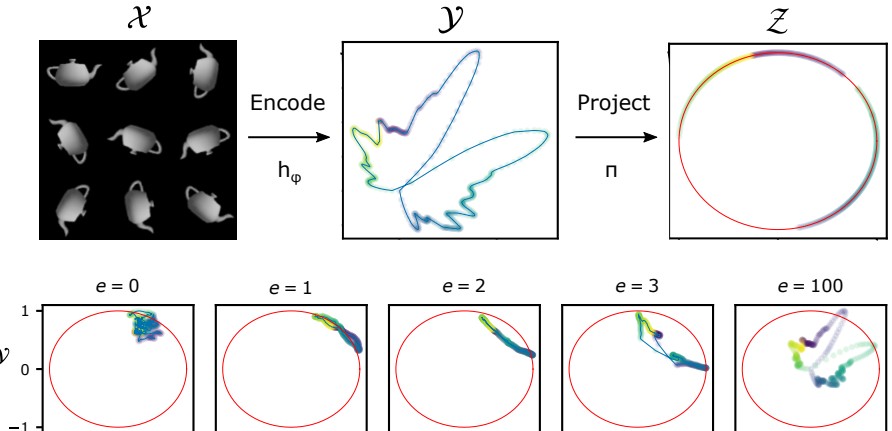

Figure 3: Homeomorphic autoencoders can encounter optimization obstructions during training. Here a network $h_\phi$ maps from data space $\mathcal{X}$ (e.g. images of rotated teapots) to an intermediate space $\mathcal{Y}$, followed by a projection $\pi$ onto a latent space $\mathcal{Z}$ with a non-trivial topology (e.g. the unit circle $S^1$). Optimization obstructions can arise when a randomly initialized network maps data onto a trajectory with topological defects, such as the crossing in a "figure 8" shape, which manifest as discontinuities after projection.

## Appendix I. Additional Figures

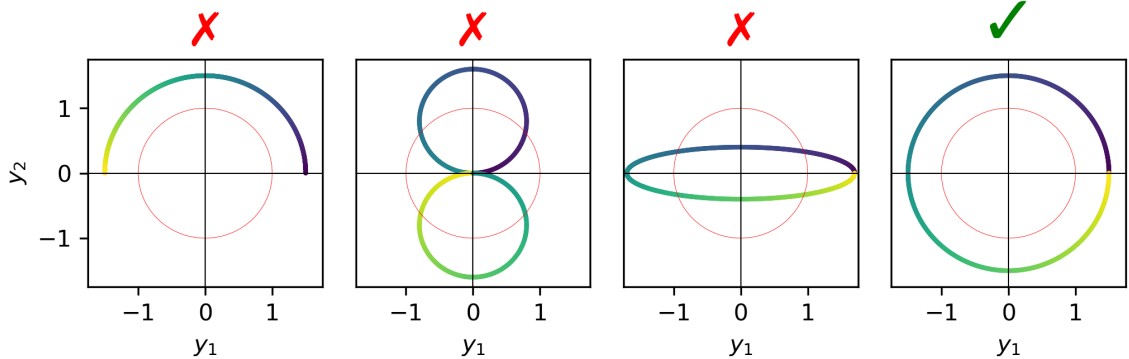

Figure 4: $\mathcal{Y}$-space of various failure cases for encoding to $SO(2)$.

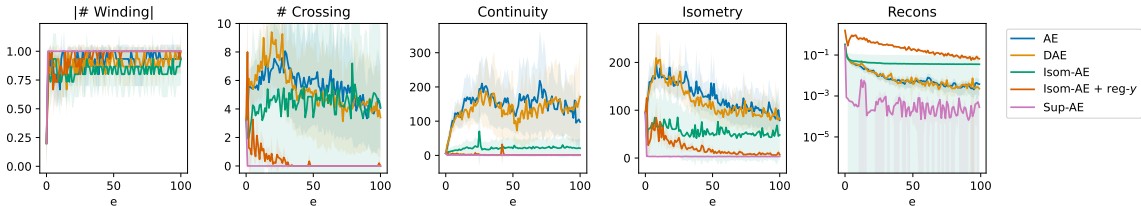

Figure 5: Learning curves of different AEs trained on the 3D-Crown dataset.

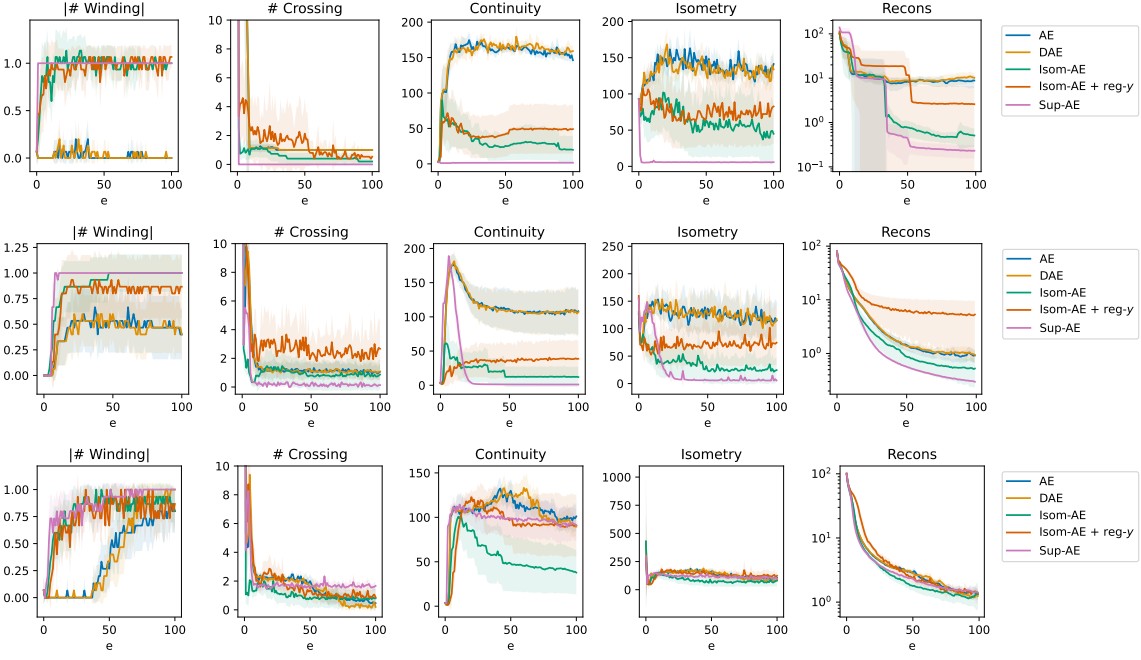

Figure 6: Learning curves of different objectives for for the Tetrominoe (*Top*), Teapot (*Middle*), and Airplane (*Bottom*) dataset.

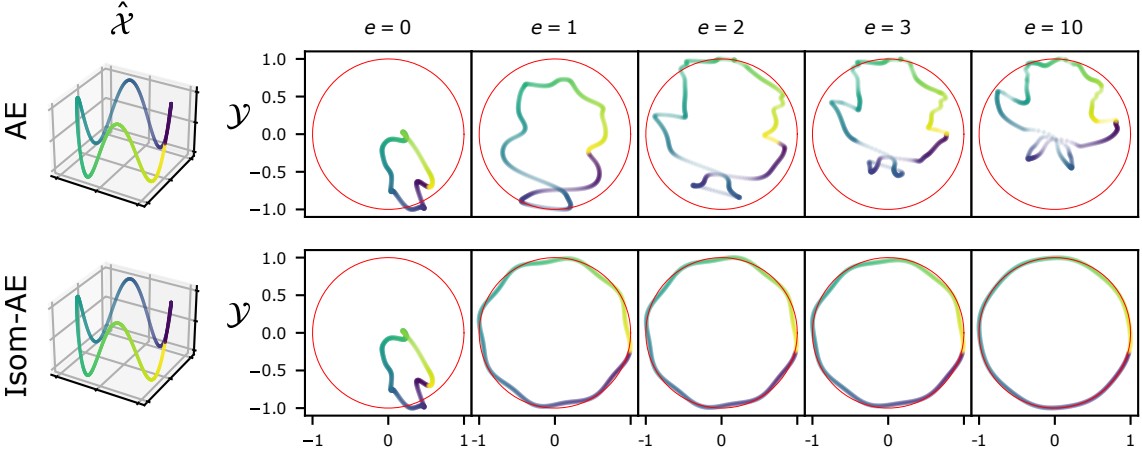

Figure 7: A comparison AE and Isom-AE objective in terms of reconstructions (*Right*) and how $\mathcal{Y}$-space evolves at different epochs (*Left*).

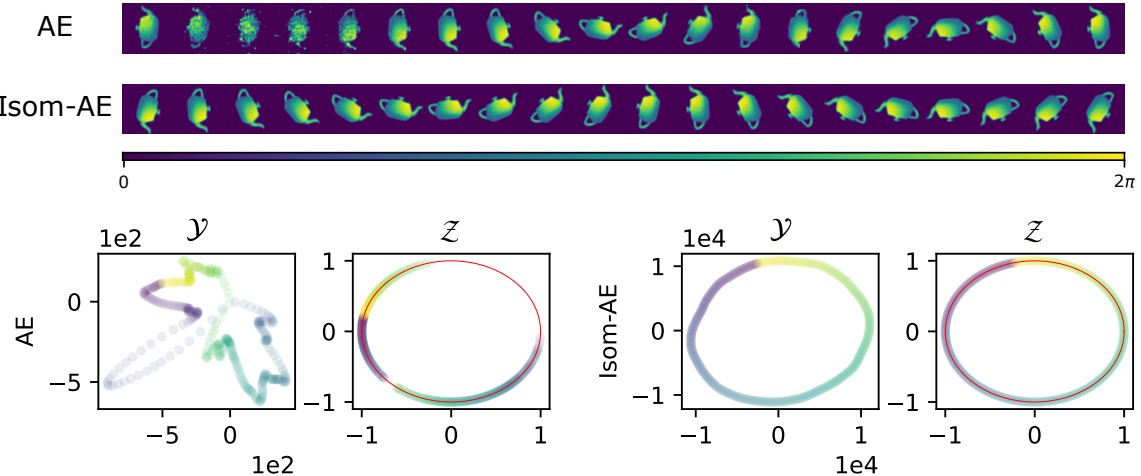

Figure 8: Latent space interpolation for the decoder (*Top*) and encoder (*Bottom*).

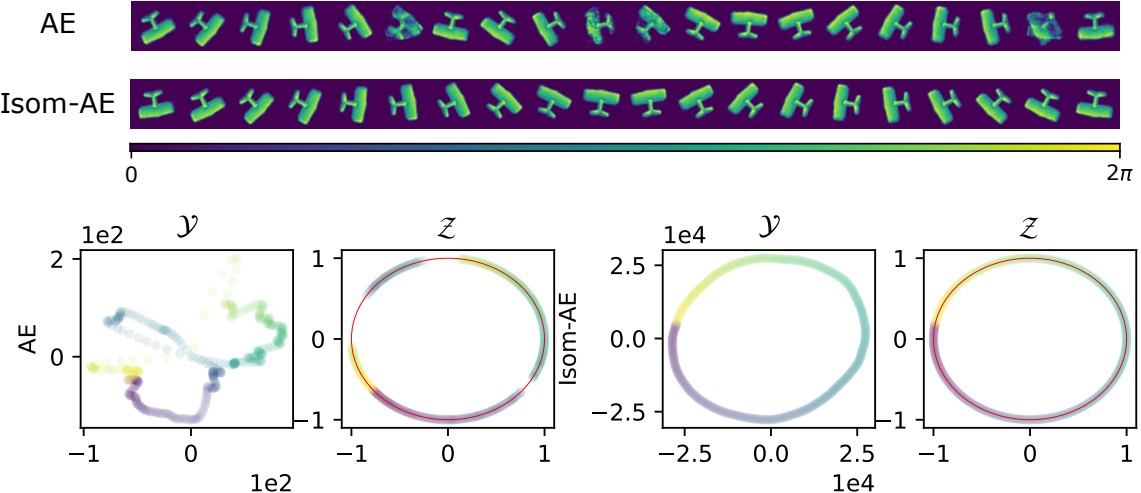

Figure 9: Latent space interpolation for the decoder (*Top*) and encoder (*Bottom*).

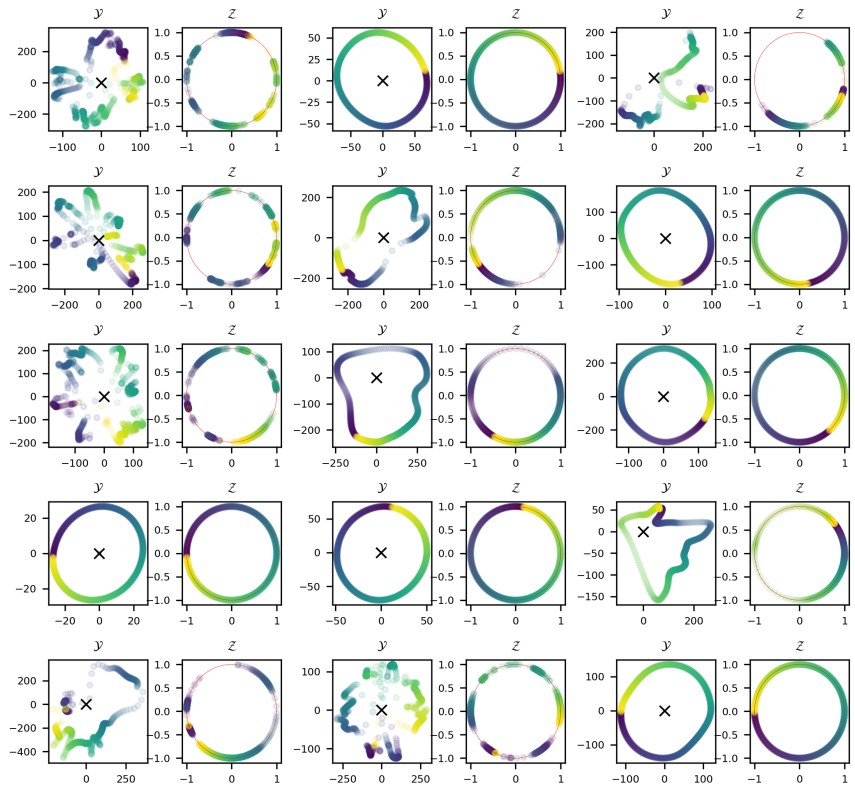

Figure 10: $\mathcal{Y}$ and $\mathcal{Z}$ space for AEs trained on the 3D-crown dataset.

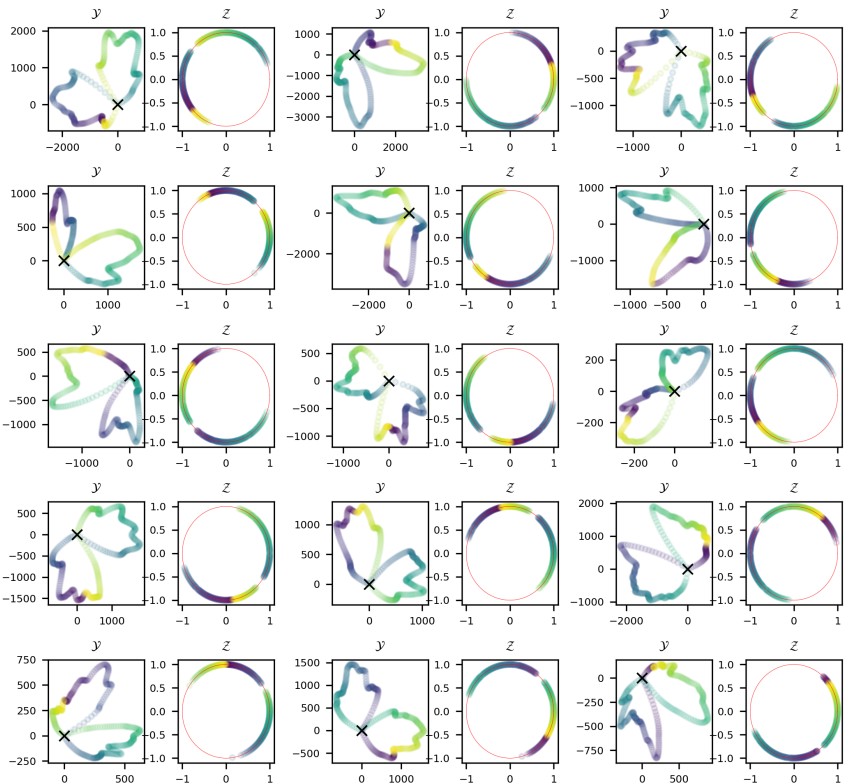

Figure 11: $\mathcal{Y}$ and $\mathcal{Z}$ space for AEs trained on the tetrominoes datasets.

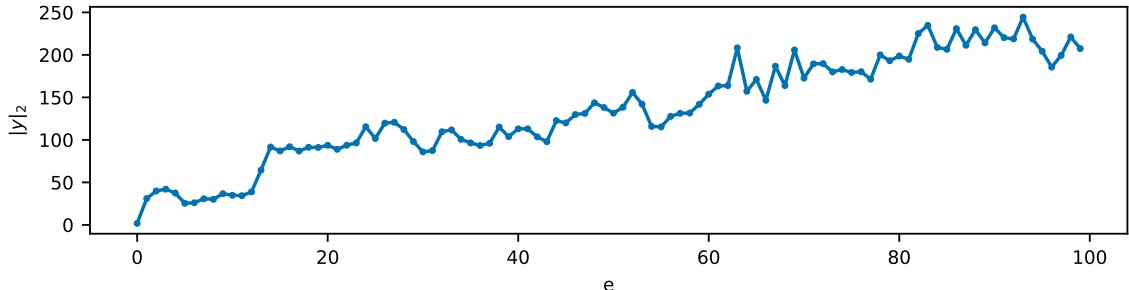

Figure 12: $\|y\|_2$ as a function of epoch for a standard autoencoder trained on teapots for seed 0.

*L-shaped Tetrominoe*

| | # H. | # W. | # C. | Cont. | Isometry | Recons |
|---|---|---|---|---|---|---|
| AE | 0/15 | 0/15 | 0/15 | $146.33 \pm 15.24$ | $141.28 \pm 39.02$ | $8.85 \pm 5.96$ |
| DAE | 0/15 | 0/15 | 0/15 | $159.24 \pm 17.29$ | $133.83 \pm 26.12$ | $10.18 \pm 5.44$ |
| Isom-AE | 12/15 | 15/15 | 12/15 | $19.77 \pm 37.48$ | $44.94 \pm 79.67$ | $0.51 \pm 0.44$ |
| Isom-AE + reg-$y$ | 9/15 | 14/15 | 9/15 | $49.07 \pm 69.49$ | $82.71 \pm 104.20$ | $2.59 \pm 7.96$ |
| Sup-AE | 15/15 | 15/15 | 15/15 | $1.60 \pm 0.05$ | $5.74 \pm 0.49$ | $0.23 \pm 0.15$ |

*Teapot*

| | # H. | # W. | # C. | Cont. | Isometry | Recons |
|---|---|---|---|---|---|---|
| AE | 2/15 | 6/15 | 5/15 | $108.35 \pm 67.06$ | $114.68 \pm 66.94$ | $0.92 \pm 0.49$ |
| DAE | 2/15 | 6/15 | 5/15 | $106.96 \pm 68.37$ | $117.22 \pm 71.50$ | $0.94 \pm 0.51$ |
| Isom-AE | 12/15 | 13/15 | 12/15 | $11.97 \pm 29.05$ | $24.48 \pm 43.99$ | $0.52 \pm 0.58$ |
| Isom-AE + reg-$y$ | 8/15 | 9/15 | 8/15 | $38.79 \pm 54.36$ | $74.51 \pm 81.42$ | $5.32 \pm 8.80$ |
| Sup-AE | 13/15 | 15/15 | 13/15 | $1.11 \pm 0.04$ | $5.67 \pm 0.48$ | $0.30 \pm 0.02$ |

*Airplane*

| | # H. | # W. | # C. | Cont. | Isometry | Recons |
|---|---|---|---|---|---|---|
| AE | 0/15 | 12/15 | 12/15 | $101.05 \pm 33.64$ | $104.94 \pm 44.70$ | $1.32 \pm 0.44$ |
| DAE | 0/15 | 15/15 | 13/15 | $89.54 \pm 23.41$ | $91.40 \pm 31.66$ | $1.24 \pm 0.54$ |
| Isom-AE | 9/15 | 12/15 | 9/15 | $37.84 \pm 49.42$ | $80.43 \pm 105.58$ | $1.35 \pm 1.19$ |
| Isom-AE + reg-$y$ | 5/15 | 13/15 | 5/15 | $91.07 \pm 70.20$ | $124.10 \pm 93.53$ | $1.40 \pm 0.71$ |
| Sup-AE | 0/15 | 15/15 | 0/15 | $90.88 \pm 35.28$ | $96.78 \pm 38.71$ | $1.41 \pm 0.21$ |

Table 5: Evaluation of different objectives trained on the Tetrominoe, Teapot, and Airplane datasets.

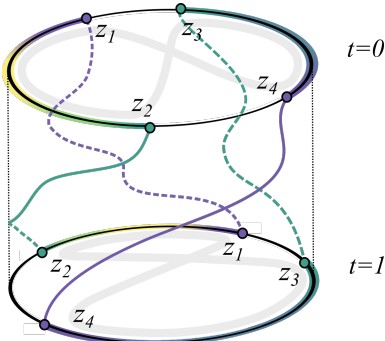

Figure 13: The figure 8 pattern in $\mathcal{Y}$ (grey) maps to two disconnected components in $\mathcal{Z}$. The cyclic order of these 4 endpoints is preserved by homotopy. Following the parameterization of the data manifold the cyclic order is $(z_1, z_2, z_4, z_3)$, which is distinct from the cyclic order of a homeomorphic embedding, either $(z_1, z_2, z_3, z_4)$ or $(z_4, z_3, z_2, z_1)$.

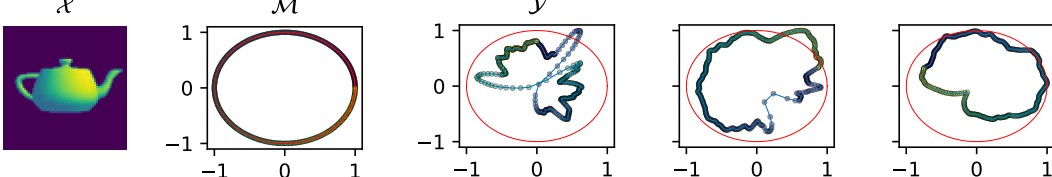

Figure 14: Intermediate space $\mathcal{Y}_x = h_\phi(\mathcal{M}_x)$ after convergence for 3 seeds.

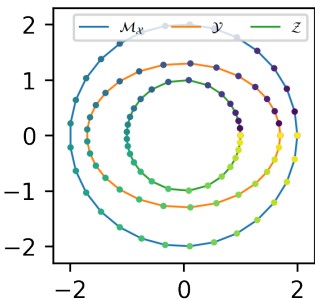

Figure 15: Example of a homeomorphism $f_\phi$ and corresponding non-isometric map $\eta_\phi$ for $\phi = (1.7, 1.3)$.

Esmaeili Walters* Zimmermann van de Meent

