# OpenReview forum: "Understanding Optimization Challenges when Encoding to Geometric Structures"
_NeurIPS.cc/2022/Workshop/NeurReps — NeurReps 2022 Poster_

### Official Review · Reviewer_rg1t · 2022-10-07
**Unclear how the theory and experiments contribute insight to field at this point**

**Confidence:** 4
**Soundness:** 2
**Presentation:** 1
**Contribution:** 2
**Overall Rating:** 3

**Summary:**

The paper develops a new method, an “isometric autoencoder,” to approximately preserve manifold distances under a composition of maps in the context of varying obstructions like a high winding number or self-intersecting manifolds.

They test this on a variety of datasets and conclude that the isometric loss function they introduce can overcome these obstructions in at least these domains, which they begin to develop some theory for. The main theoretical contribution is that the figure 8 embedding cannot be mapped via homeomorphism and that the growing magnitude of the embeddings makes learning more difficult.


**Questions:**

-Table 1 is very confusing. No definition is given for #H or isometry, and no definitions (even high-level ones or brief descriptions) are given for the model names anywhere. One has to dig into page 16 in the appendix to glean what these results mean or why they are even significant. If the appendix is deleted, I cannot tell what this means. Even high-level would suffice (e.g. “Isometric error” in the title). You should not assume that the reader will look at the appendix--results should be self-contained within the paper and then expanded upon, time permitting, in the appendix.

-Definition 2 (no definition 1, either?): mathematical definitions are typically reserved for descriptions of mathematical objects that satisfy certain criteria/properties. This is not a definition but a subjective notion of “optimality” that the authors hope to accomplish (a fair goal nonetheless, but not a definition). A definition might define optimality in terms of mean-square error or isometric loss and then further proofs might show that your method achieves this lower bound. A common theme throughout the paper are definitions and propositions of little to unclear rigor or significance.

-Why are the vanilla AE and DAE architectures relevant baselines of comparison? Appendix A examines related work on topological constructions, and there is an abundance of literature on equivariance in neural networks (see Group Equivariant Convolutional Networks by Cohen, Welling 2016 since you use convolutional structure). Are there more recent, relevant updates rather than architectures that we expect not to be endowed with this isometric property?

-The main proposition in the text (proposition 1 in the text, also called proposition 7 in the appendix. Another side note is that many propositions seem to overlap or are literal repeats with different numbers which is confusing) is a one-line (perhaps even trivial?)  proof. What insight does it bring to the paper? Why is it the featured proposition? Proposition 6 seems much more involved and more central to section 3.1 (and more innovative) than Proposition 1.  Why haven't you pointed us to this result in the main text?

-Proposition 4 is a very specific, rigorous claim left to a very non-rigorous figure that, personally, I find hard to relate to your claim. If you have the proof and the insight, can you share this under your proposition? Can you guide the reader more, even if only intuitively? At the very least, can you group Figure 13 with the proposition for coherency?

-Proposition 4 is very similar to proposition 6, so perhaps this is where the meat lies. Can you tie these two propositions together and help us understand why they should be differently named propositions?

-I’m not familiar with many of the objects you use in Proposition 6, but have some comments about this proof. First, on a non-technical level, it is beyond hard to read. Sets bleed over from line to line and notation is used at the expense of words and insight. Using the align or equation environment in TeX is a must here: this is a wall of (very dense) text. Crucial steps are hand-waved without sufficient support. Why does “linear interpolation show that $D$ is connected”? Can you show this? How are you constructing the interpolation? You claim $\sigma$ reverses the order of at least two elements but the alleged sign change is not apparent to me: why can you use the intermediate value theorem here?

-Why are Proposition 5 and 9 different?

-Notation frequently jumps around, is left undefined, or is abstracted at the expense of clarity of what you actually did. This is very tiring for the reader and detracts from our ability to evaluate the text. Equation 5, for example, (which is mentioned several times throughout the paper) uses a very abstract notion of distance when you’re using $\ell_2$ in all experiments. Why not just write this with the $\ell_2$ loss and justify its presence as meaningful? Much of the notation is completely undefined or left for the reader to figure out for themselves. What are $\phi$ and $\theta$ in the problem statement? Can you say that these are the parameters of the network instead of leaving us to find this later? In the theory section, where do $y$ and $z$ come from and why do you need to write them here? Why not just say “unit sphere” instead of $S_1$, or that we project points from the unit sphere to the latent space using a Gram-Schmidt procedure?  Where does $g$ come from in the definition of winding number? Minimal, consistent notation helps the reader follow along with the core results of your paper, but the writing often introduces unnecessary jargon which obfuscates the results.


**Limitations:**

The authors state that the limitations relate to the idealized nature of the theoretical assumptions and the simplified definition of isometry. At a high level, however, I find neither of these particularly concerning as you can’t be expected to generalize all of your results, especially for an early-stage extended abstract. Thorough investigation of toy models is more than sufficient. Almost-isometries are well defined, too, for metric spaces regardless of dimension. Much more concerning to me would be the variance in the main results presented in Figure 1. Isom-AE seems highly unstable with respect to the given definition of isometry, and this is never mentioned in the paper. Why is the range so consistently large? Why does regularization seem to make the range even larger? How many trials did you run? Is #H as unstable as the measure of isometry? Also, see earlier comments about baseline comparisons.


**Recommended Decision:**

2: Borderline

**Relevance:**

4: Highly relevant

**Strengths And Weaknesses:**

The paper presents an array of theoretical and experimental results to explain the algorithmic properties in detail, so this perspective and approach is a huge plus for its potential to be expanded upon in future work.

However, the paper is very hard to read and piece together the actual contributions other than what’s claimed in the text. Many key results in the meat of the paper, not the appendix, are left not only unexplained in terms of significance but even definition, leaving the reader to wonder what the table or theoretical result is referring to (see points below for more details on this point). If the appendix were deleted, the paper would (literally) be incomprehensible. While it’s admirable to include as much content and experiments as possible, I would’ve much preferred less experiments for more discussion, even if only at a high-level and especially for 23 pages of a 4 page abstract (why not submit a proceedings version?). The paper itself is very unpolished and could benefit from several revisions and future drafts.

Proofs and definitions are often used either incorrectly in motivation, not clearly used in further arguments, or not explicitly proved. Proposition 6 is the most rigorous, innovative proof but isn’t cited and is hardly mentioned in the main text. See more details below on specific propositions. Reproducing proofs from StackExchange (while good to cite and certainly in no way not able to be used when working on results) detracts from the professional quality of the document and the perceived insight of the theoretical results. I'm unsure why the magnitudes of the embeddings are a novel problem: instead of penalizing this with regularization that you need to tune, why not just normalize this after each gradient update a la batch norm or projected gradient descent?

As expanded upon below, the experimental results are also of questionable significance. It’s not clear to me why AE and DAE are relevant baselines and how comparing to these models actually builds upon key themes in equivariance, group-theoretic or geometric learning. At the very least, it seems that your encoders could have incorporated some notion of local preservation of distance (geodesic, $\ell_2$ etc.) rather than a vanilla architecture and loss function since such work is so old (~2006 or earlier). Without this information, it’s hard to tell whether or not these results are promising (see my comments on result variance below, too). The loss function for Isom-AE strikes me as conceptually similar to locally linear embedding, which should handle these relatively simple domains quite well. There are other works, older works, that try to preserve certain geometric properties like isometry: there should be a comparison to these methods rather than generic architectures that aren't specialized to geometry. I see this as the central weakness of the current version (besides the issues with structure and organization).


**Submission Track:**

Extended Abstract (4 Page)

---

### Official Review · Reviewer_nzXw · 2022-10-15
**Highly relevant work, maybe too strong on the theoretical definition**

**Confidence:** 2
**Soundness:** 3
**Presentation:** 3
**Contribution:** 3
**Overall Rating:** 7

**Summary:**

This work studies the optimization challenges that arise when guiding the shape of latent space to a given geometric structure when training autoencoders. In particular, the focus is on the following issues: singularity, winding number and the non-isometry that can arise even when the resulting embeddings are homeomorphic. A method (isometric autoencoders) is proposed to improve stability and convergence with respect to these issues.

**Questions:**

Have the authors tried different ways to weigh the *y regularization*? The results in Table 1 show that it is not helping the model at all, so I'm curious about its impact when weighted down.

**Limitations:**

The authors already addressed the limitations of the work.

I want to point out this work by Gropp et al (https://arxiv.org/abs/2006.09289). where, to the best of my knowledge, the "isometric autoencoders" were first introduced by means of a latent regularizer.

**Recommended Decision:**

3: Accept

**Relevance:**

4: Highly relevant

**Strengths And Weaknesses:**

From my point of view, this work's theoretical perspective is both a good strength and a fair weakness.

The content is soundly described thanks to the mathematical formulation, but this makes its fruition much heavier even for members of this community. However, the work is well structured, and motivated, and brings sound results of interest to the community.

Personally, I was not able to follow central parts of this paper (and that is why I'm giving a low confidence score and why I cannot give an "Excellent" rating to Presentation and Soundness), but the overall picture is quite clear.

**Submission Track:**

Extended Abstract (4 Page)

---

### Official Review · Reviewer_4AYy · 2022-10-18
**Understanding optimization challenges when encoding to geometric structure**

**Confidence:** 3
**Soundness:** 3
**Presentation:** 2
**Contribution:** 3
**Overall Rating:** 7

**Summary:**

This paper suggests an approach to representation learning by using inductive biases, especially those that rely on geometric priors. The proposed approach is claimed to make more accessible how to reason about similarities of different instances of the data. Yet experience
has shown that the use of inductive biases may give rise to optimization challenges. Hence based on theoretical grounds the manuscript suggests using an autoencoder objective that maps equidistant points on the abstract manifold to equidistance points on the latent space.
The manuscript also includes results from the testing of the suggested approach.

**Questions:**

1. It is suggested for the authors to clarify what claims are indeed supported by theory and which ones are inferred from the simulations and experiments that the authors have performed.  A lot of material is presented in the  Appendices  but if indeed the appendices will be
published together with the manuscript (i.e., they are quite long and contain much more text than the manuscript) it will be very helpful if the reader can be directed on what claim is presented in what appendix.

2.  The appendices themselves are not very well presented.  The authors should shorten them and decide which appendices are more important and which are less. Also, many figures and tables are presented in the appendices without explanations. The appendices need more work in terms of clarity, language, removing typo mistakes, etc.


**Limitations:**

There is not a sufficient discussion of the limitations of the work. Such discussion should be added.


**Recommended Decision:**

2: Borderline

**Relevance:**

3: Solid fit

**Strengths And Weaknesses:**

Originality:   the work is novel and quite exciting and shows a potential for suggesting how to incorporate geometric priors into the learning of optimal representation when encoding geometric structure.  It also makes mathematically sound analyses about distances between data representation on the abstract versus the latent space.
Quality: the claims seem technically sound and are backed by theoretical reasoning. It is difficult to assess, however, whether the claims made are indeed supported because it is based on "experimental" testing and it is difficult to assess based on the tested instances how general these claims are.  Clarity:  this is an extended abstract, but most of the theoretical analysis is presented in numerous appendices that need to be delivered more concisely and written more clearly. Also, the appendices are not very well organized mixing technical details and theory without sufficient explanations.
Significance:  The results can be of interest to the community.


**Submission Track:**

Proceedings Paper (9 Page)

---

### Decision · Program_Chairs · 2022-10-21

Accept (Poster)